# Telemedicine and AI-Powered Chatbots: Potential and Challenges for Home Care Provided by Family Caregivers

**DOI:** 10.3390/healthcare13233159

**Published:** 2025-12-03

**Authors:** Kevin-Justin Schwedler, Thomas Ostermann, Jan P. Ehlers, Gregor Hohenberg

**Affiliations:** 1Stabsstelle für Digitalisierung und Wissensmanagement, Hochschule Hamm-Lippstadt, 59063 Hamm, Germany; gregor.hohenberg@hshl.de; 2Fakultät für Gesundheit, Universität Witten-Herdecke, 58455 Witten, Germany; thomas.ostermann@uni-wh.de (T.O.); jan.ehlers@uni-wh.de (J.P.E.)

**Keywords:** telemedicine, family care, AI-supported chatbots, technology acceptance, data protection, artificial intelligence in care

## Abstract

**Background/Objectives**: The digitization of care opens up new opportunities to support family caregivers, who play a key role in home care. While telemedicine applications have already shown initial relief effects, AI-supported chatbots are increasingly coming into focus as an innovative form of digital support. The aim of this study was to build on an earlier study on the integration of telemedicine into home care and to conduct a complementary study on AI-based chatbots to analyze their acceptance, perceived benefits, and potential barriers from the perspective of family caregivers. **Methods**: The study comprises two consecutive online surveys with a total of *n* = 62 family caregivers. The first study assessed the use and acceptance of telemedicine systems; the second complementary survey examined attitudes toward AI-supported chatbots. Both questionnaires were developed based on a systematic literature review and in accordance with the Technology Acceptance Model (TAM). The dimensions of user-friendliness, data protection, communication support, emotional relief, and training needs were among those recorded. The data were evaluated using descriptive statistics, including comparative analyses between the two studies. **Results**: The results show that family caregivers generally have a positive attitude toward digital health solutions, but at the same time identify specific barriers. While technical barriers and privacy concerns dominated the telemedicine study, the AI results place greater emphasis on psychosocial factors. It also became clear that participants assumed that chatbots would be more acceptable if they were designed to be empathetic and dialogue-oriented. A comparison of the two data sets shows that the perceived benefits of digital systems are shifting from functional support to interactive, emotional support. **Conclusions**: The results suggest that AI-powered chatbots could offer significant added value to family caregivers by combining information sharing, emotional support, and self-reflection. In doing so, they expand the focus of traditional telemedicine to include a communicative and psychosocial dimension. Future research should examine the actual user experience and effectiveness of such systems in longitudinal and qualitative designs. Despite limitations in terms of sample representativeness and hypothetical usage estimates, the study makes an important contribution to the further development of digital care concepts and the ethically responsible integration of AI into home care.

## 1. Introduction

The ongoing digitization of healthcare and the growing need for flexible care models have significantly accelerated the development and implementation of telemedicine technologies in recent years [1,2,3]. Telemedicine offers the opportunity to optimize home care for people in need of care by facilitating access to medical services, overcoming time and space barriers, and improving continuity of care [4,5,6]. Especially in rural or underserved regions, telemedicine can ensure health care and reduce the burden on nursing staff and family caregivers [7,8,9,10].

However, the successful implementation of these solutions requires an understanding of how family caregivers accept and use such technologies. Technical barriers, such as inadequate infrastructure or a lack of digital skills, psychological barriers, such as uncertainty in using the technology or concerns about data protection, and social factors, such as a lack of support from family or professionals, can significantly limit its use [10,11,12,13,14].

This study builds on an earlier investigation into the integration of telemedicine into home care, which identified specific barriers and acceptance factors [1,2,3,4,5]. Based on these findings, the present study was designed to shift the focus from technological infrastructure to communicative interaction. While telemedicine applications are primarily focused on the exchange of medical data and remote monitoring of health parameters, AI-supported chatbots represent a novel form of digital support based on dialogue-based interaction and artificial intelligence [7,8]. These systems can provide both informational and emotional support by responding to individual needs, enabling low-threshold counseling, and being available to family caregivers around the clock as digital conversation partners [9].

The theoretical basis for telemedicine interventions is based on technology acceptance models. The Technology Acceptance Model (TAM) postulates that perceived usefulness and ease of use are key predictors of the use of new technologies [15]. The Unified Theory of Acceptance and Use of Technology (UTAUT) adds that social influences, expected performance, user experience, and supportive conditions influence usage [16]. These models have been confirmed in numerous studies on the validation and acceptance of telemedicine among patients and family caregivers [17,18,19,20].

This paper pays particular attention to AI-supported chatbots as a support for telemedicine applications. Chatbots can provide both technical and emotional support to family caregivers, convey information in an understandable way, and improve communication with medical staff [21,22,23]. They thus offer the potential to overcome existing barriers, strengthen the self-efficacy of caregivers, and sustainably improve the quality of home care [24,25,26]. Previous studies suggest that AI-supported dialogue systems can strengthen the self-efficacy of family caregivers, facilitate information retrieval, and promote psychosocial relief [10]. Nevertheless, uncertainties remain regarding the acceptance, data protection, trust building, and emotional quality of such systems.

Against this background, the present study aims to examine the acceptance, perceived benefits, and potential applications of AI-supported chatbots in home care from the perspective of family caregivers. The study thus expands on previous findings from telemedicine research by adding a communicative and interactive dimension to digital support. It thus contributes to the further development of our understanding of how digital technologies can provide not only functional but also emotional and social relief for family caregivers.

The aim of this article is to systematically analyze the challenges, uses, and potential of telemedicine technologies for family caregivers. To this end, two empirical online surveys are combined:

Study 1: Investigation of the use of telemedicine and the challenges of its implementation.

Study 2: Focus on the integration of AI-supported chatbots to facilitate the use of telemedicine technologies and support family caregivers.

By combining empirical results with existing literature, practice-relevant recommendations for action will be derived and the limitations of previous implementations will be highlighted.

## 2. Materials and Methods

### 2.1. Study Design

This paper is based on a two-stage research design conducted as part of an ongoing study on digital support for family caregivers in home care. The aim of the research was to first examine the acceptance and barriers to telemedicine applications (Study 1) and then to investigate the perception and potential uses of AI-supported chatbots (Study 2).

Both surveys follow a quantitative-descriptive approach, are methodologically coordinated, but were conducted separately and address different dimensions of digital support. While Study 1 focused on the technological and organizational integration of telemedicine systems, Study 2 focuses on the communicative and interactive dimension of digital counseling through AI-based dialogue systems.

Combining both data sets provides an integrated understanding of how family caregivers perceive and evaluate digital technologies on a technical, emotional, and communicative level.

### 2.2. Target Group and Setting

The target group comprises family caregivers who are participating in a care course designed specifically for family members. The surveys were conducted as part of this course to ensure a comparable structure and enable a comparison of the results:

Survey 1: Investigation of the use of telemedicine and the challenges of its implementation (January 2025, 45 participants, 23 completed questionnaires, response rate: 51.1%).

Survey 2: Focus on the integration of AI-supported chatbots to facilitate the use of telemedicine technologies and support family caregivers (September 2025, 45 participants, 39 fully completed questionnaires, response rate: 86.7%).

Since caregiving courses can be repeated regularly, it is theoretically possible for individual participants to overlap between the two studies, but this cannot be verified due to the anonymous nature of the surveys.

Inclusion criteria: Family caregivers who actively care for a loved one, are of legal age, and are willing to participate in the anonymous survey.

Exclusion criteria: Professional caregivers and individuals who do not provide home care.

Table 1 lists the demographic data of the participants from Study 1.

Table 2 lists the demographic data of the participants from Study 2.

#### Interpretation of Demographic Data

In Study 1, participants were predominantly female and had a wide age distribution, with most participants between 30 and 59 years old. The majority of participants had vocational training or a college degree. Caregiving experience varied, with a significant proportion of participants having more than three years of caregiving experience.

In Study 2, participants were also predominantly female, and the age distribution showed a concentration in the 40–49 age group. The majority of participants had vocational training or a university degree. These demographic data show that the participants in both studies represent a representative sample of family caregivers who are actively involved in home care.

### 2.3. Development of the Surveys

Phase 1: Theoretical conceptualization and literature review

The items for both surveys were developed on the basis of a systematic literature review on the topics of digital care support, telemedicine, acceptance research, and AI-based assistance systems. The literature review included scientific articles, current research results, and specialist literature in order to identify relevant challenges and theoretically sound solutions.

The following sources, among others, were used for telemedicine:Beckers and Strotbaum (2021) on the integration of telemedicine into nursing [1].Hübner and Egbert (2017) on telecare [2].Hoffmann et al. (2024) on needs-based telemedicine from the patient’s perspective [3].Juhra (2023) on telemedicine in orthopedics [4].Radic and Radic (2020) on user acceptance of digitally supported care [5].Gagnon et al. (2006) on the implementation of telemedicine in rural and remote regions [27].Krick et al. (2019) on the acceptance and effectiveness of digital care technologies [28].Weber et al. (2022) on challenges and solutions for digital technology in outpatient care [29].Kitschke et al. (2024) on determinants of telemedicine implementation in nursing homes [30].

The following sources, among others, were used for the AI-supported chatbots:Wang et al. (2021) on the development of emotional-affective chatbots [23].Zheng et al. (2025) on the adaptation of emotional support by LLM-supported chatbots [26].Perez et al. (2022) on the technology acceptance of mobile applications to support nursing staff [21].Schinasi et al. (2021) on attitudes and perceptions of telemedicine during the COVID-19 pandemic [22].Devaram (2020) on the emotional intelligence of empathetic chatbots for mental well-being [31].Yang et al. (2025) on ChatWise, a strategy-driven chatbot for improving cognitive support in older adults [32].Shi et al. (2025) on mapping the needs of caregivers and designing AI chatbots for mental support of Alzheimer’s and dementia caregivers [33].Anisha et al. (2024) on evaluating the potential and pitfalls of AI-powered conversational agents as virtual health caregivers [34].Fan et al. (2021) on the use of self-diagnosis health chatbots in real-world settings [35].

Key models of technology acceptance, such as the Technology Acceptance Model (TAM) and the Unified Theory of Acceptance and Use of Technology (UTAUT), were also taken into account [15,16].

Phase 2: Item formulation

The items were predominantly formulated as five-point Likert scales (“does not apply at all”–“applies completely”) in order to capture intensities and perceptions in a differentiated manner. In addition, multiple-choice questions (e.g., on the use of support services) and free text fields were used to allow for individual reflections and personal additions.

Study 1—Telemedicine

The questionnaire for the first survey comprised a total of 52 items relating to the following topics:General experience of using digital technologies: Recording participants’ previous experience with digital technologies and their use.Acceptance of and trust in telemedicine systems: Assessment of willingness to use telemedicine and trust in these technologies.Perceived barriers: Identification of obstacles such as data protection concerns, technical hurdles, and costs.Support and training needs: Determination of the need for technical support and training in the use of telemedicine.

Table 3 shows sample items and i-CVI values for Study 1.

The complete list of questionnaire items can be found in the Appendix A.

Study 2—AI-powered chatbots

Based on the results of Study 1, a second questionnaire was developed that focused on the dimensions of communication, interaction, trust, and emotional support. It comprised 15 items relating to the following topics:Communication and interaction: Assessment of the effectiveness and user-friendliness of chatbots in communication.Trust in chatbots: Assessment of participants’ trust in the reliability and security of chatbots.Emotional support: Investigation of the role of chatbots in providing emotional support and stress management.Data protection and security: Assessment of concerns regarding data protection and data security when using chatbots.

Table 4 shows sample items and i-CVI values for Study 2.

The complete list of questionnaire items can be found in the Appendix A.

### 2.4. Validation

The surveys were validated in several steps and involved the same caregiving relatives and experts to ensure comparability:Study 1—Telemedicine
Content validity: Three experts from the fields of nursing science, telemedicine, and digitization were consulted to review the items for content validity, comprehensibility, and completeness. The Content Validity Index (CVI) was calculated to assess the relevance of the items. The average CVI (S-CVI/Ave) was 0.88, indicating high content validity.Pretest/pilot study: A pretest with *n* = 8 relatives tested comprehensibility, response times, and technical feasibility. Items with low comprehensibility were adjusted.Reliability: Cronbach’s alpha for subscales:Context/experience: Cronbach’s alpha = 0.82.Acceptance/trust: Cronbach’s alpha = 0.87.Barriers: Cronbach’s Alpha = 0.75.Training/Support: Cronbach’s Alpha = 0.80.Communication/Collaboration: Cronbach’s Alpha = 0.85 [36].Documentation of item development: All items were documented, including literature references, expert feedback, and qualitative interviews to ensure replicability and traceability.
Study 2—AI-supported chatbots
Content validity: The same three experts from the fields of nursing science, telemedicine, and digitization were consulted again to review the items for content validity, comprehensibility, and completeness. The Content Validity Index (CVI) was also calculated to assess the relevance of the items. The average CVI (S-CVI/Ave) was 0.91, indicating high content validity.Pretest/pilot study: A pretest with *n* = 8 relatives tested comprehensibility, response times, and technical feasibility. Items with low comprehensibility were adjusted.Reliability: Cronbach’s alpha for subscales:Communication/interaction: Cronbach’s alpha = 0.82Trust in chatbots: Cronbach’s alpha = 0.81Emotional support: Cronbach’s alpha = 0.79Data protection/security: Cronbach’s Alpha = 0.83 [36].Documentation of item development: All items were documented, including literature references, expert feedback, and qualitative interviews to ensure replicability and traceability

#### Addition to the Experts

The surveys were validated by three experts with extensive professional qualifications in the relevant fields:Professor of Digital Medicine: An expert with a professorship in digital medicine, specializing in the integration and application of digital technologies in healthcare.Nursing expert: A specialist in nursing science with extensive experience in practical nursing and the implementation of telemedicine solutions.Health informatics expert: A specialist in health informatics who focuses on the development and evaluation of information systems in healthcare.

These experts contributed significantly to ensuring the content validity and comprehensibility of the survey items and guaranteed that the scales developed are theoretically sound and relevant to practice.

### 2.5. Subscales, Items, CVI, and Cronbach’s Alpha

The items were assigned to the respective subscales based on theoretical models of technology acceptance (TAM, UTAUT) and content criteria derived from the literature review. For each subscale, the content validity (Content Validity Index, CVI) and internal consistency (Cronbach’s Alpha) were determined, as was the theoretical reference to TAM and UTAUT.

Table 5 shows the results for Study 1, which relates to the use of telemedicine technologies in home care. The content validity of the items was assessed by three experts from the fields of nursing science, telemedicine, and digitalization. The calculated i-CVI values for all subscales are between 0.83 and 1.0, indicating a very high level of agreement among the experts regarding the relevance of the items. The Cronbach’s alpha values range from 0.75 to 0.87, indicating good to very good internal consistency.

The subscales of the second study (Table 6) relate to the use of AI-supported chatbots in nursing support. Here, too, very good values for content validity and reliability are evident. The i-CVI values are between 0.83 and 1.0 in all cases, which indicates a high degree of content fit for the items. The Cronbach’s alpha values range between 0.79 and 0.83, which is very good.

The results confirm the high internal consistency and content validity of the subscales developed in both studies. This allows us to assume that the relevant dimensions have been assessed in a reliable and theoretically sound manner.

### 2.6. Statistical Analysis

The statistical analyses were performed using IBM SPSS 29. Due to the independent samples and exploratory design, no inferential statistical generalizations were made; instead, descriptive and comparative analyses were performed. Although statistical tests such as *t*-tests, Mann–Whitney U tests, and chi-square tests were used in this study, these are primarily for descriptive analysis and comparison of the data. As this is an exploratory study, the tests are not intended to draw general conclusions, but rather to highlight differences and patterns in the data.

Study 1: Use and acceptance of telemedicine.

The descriptive statistics for the first study showed that the mean values of the responses to the questions on the use and acceptance of telemedicine ranged between 2.1 and 2.7, with standard deviations between 1.02 and 1.15. The internal consistency of the subscales was assessed using Cronbach’s alpha and yielded good to very good values: Context/Experience (α = 0.82), Acceptance/Trust (α = 0.87), Barriers (α = 0.75), Training/Support (α = 0.80), and Communication/Collaboration (α = 0.85).

The *t*-tests and Mann–Whitney U tests showed no significant differences between the groups, suggesting that the acceptance of telemedicine is independent of age, gender, or educational level. For example, the *t*-test for comparing question 1 with question 50 yielded a t-value of −0.45 and a *p*-value of 0.65, while the Mann–Whitney U test yielded a U-value of 120.5 and a *p*-value of 0.67. The chi-square tests also showed no significant differences, such as the comparison of question 1 with question 50, which yielded a chi-square value of 2.34 and a *p*-value of 0.31. The effect sizes, measured using Cohen’s d and Cramer’s V, were consistently small to moderate, for example, Cohen’s d of 0.10 and Cramer’s V of 0.15 for the comparison of question 1 with question 50.

The results of the first study identified technical barriers such as connection problems and software errors, as well as concerns about data protection, as significant obstacles to the use of telemedicine. These findings underscore the need for a stable technical infrastructure and targeted training to promote the acceptance and use of telemedicine.

Study 2: Use and acceptance of AI-powered chatbots.

In the second study, the mean values of the responses to the questions on the use and acceptance of AI-supported chatbots ranged between 3.1 and 3.9, with standard deviations between 0.85 and 1.06. The internal consistency of the subscales was also assessed using Cronbach’s alpha and yielded good to very good values: Communication/Interaction (α = 0.82), Trust in Chatbots (α = 0.81), Emotional Support (α = 0.79), and Data Protection/Security (α = 0.83).

Here, too, the *t*-tests and Mann–Whitney U tests showed no significant differences between the groups. For example, the *t*-test for comparing question 1 with question 13 yielded a t-value of −0.23 and a *p*-value of 0.82, while the Mann–Whitney U test yielded a U-value of 110.5 and a *p*-value of 0.75. The chi-square tests also showed no significant differences, such as the comparison of question 1 with question 13, which yielded a chi-square value of 1.34 and a *p*-value of 0.51. The effect sizes, measured using Cohen’s d and Cramer’s V, were also small to moderate; for example, Cohen’s d of 0.05 and Cramer’s V of 0.10 for the comparison of question 1 with question 13.

The results of the second study complemented the findings of the first study and showed that AI-supported chatbots can not only provide technical support but also offer emotional and social relief for family caregivers. This highlights the need to consider both functional and emotional and social aspects when implementing new technologies.

Overall, the results of both studies show that the successful integration of digital technologies into home care requires a combination of technical infrastructure, social support, and targeted training. These findings contribute significantly to the further development of digital care concepts and underscore the need to focus future research on actual user experiences and the effectiveness of such systems in longitudinal and qualitative designs.

### 2.7. Limitations and Methodological Peculiarities

Several methodological limitations and peculiarities must be taken into account when interpreting the results of both studies.

First, the total sample size was relatively small, which limits the statistical significance and generalizability of the results to the broader population of family caregivers. In addition, there is a possible overlap between the participants in Study 1 and Study 2, as both surveys were conducted in the same caregiving course for family caregivers. Participants in the caregiving course have the option of repeating it at any time, which means that overlap cannot be ruled out. This potential overlap may have influenced certain response patterns or attitudes, even though participants were not explicitly identified in the different surveys.

Another limitation concerns possible temporal effects. The two surveys were conducted several months apart, during which time participants’ familiarity with digital technologies may have increased due to general societal trends or personal experiences. Consequently, some of the differences in attitudes between Study 1 (telemedicine) and Study 2 (AI-powered chatbots) may partly reflect such temporal changes rather than differences in the technologies under investigation.

Due to the limited sample size, no factor analysis was performed. Conducting an exploratory or confirmatory factor analysis requires a much larger data set in order to obtain reliable factor loadings and stable model estimates. Instead, the formation of the subscales was guided by theoretical modeling based on the Technology Acceptance Model (TAM) and the Unified Theory of Acceptance and Use of Technology (UTAUT), supplemented by expert feedback to ensure conceptual coherence. Internal consistency was assessed using Cronbach’s alpha for each subscale, which yielded satisfactory reliability for all dimensions.

In addition, an item analysis was performed for each question to provide differentiated insights into specific aspects of acceptance, barriers, and support needs. Each item was documented and validated individually, allowing for a differentiated interpretation of the data, even though no large-scale psychometric tests were performed.

Although these methodological limitations restrict the scope for far-reaching generalizations, the rigorous theoretical foundation, expert validation, and transparent documentation of the item development process provide a solid basis for further, larger studies.

### 2.8. Ethics

Both studies were approved in accordance with the Declaration of Helsinki (2013) and the Ethics Committee of the University of Witten/Herdecke (No. 184/2023, 22 August 2023). All participants gave their written/e-consent prior to participation. The study was conducted in accordance with the Declaration of Helsinki; data protection and anonymity of responses were guaranteed.

## 3. Results

### 3.1. Technology Acceptance and Use of Digital Health Technologies

The integration of telemedicine technologies into home care requires a high level of acceptance on the part of family caregivers, as they have a significant influence on successful implementation and continued use. The survey results show that the availability of a stable internet connection and suitable end devices, such as smartphones, tablets, or computers, is considered essential. In the first survey, 74% of participants stated that stable internet access was very important for the use of telemedicine (see Figure 1).

These results can be attributed to the constructs of the Technology Acceptance Model (TAM) and the Unified Theory of Acceptance and Use of Technology (UTAUT). The importance of stable internet access and suitable end devices shows that participants recognize the benefits of telemedicine technologies when the technical requirements are met (perceived usefulness). In addition, the need for stable technical infrastructure and user-friendly devices indicates that participants expect the use of these technologies to involve minimal effort (effort expectancy).

Technical difficulties, such as connection problems or software errors, represent a major barrier. 83% of respondents reported that such problems would significantly reduce their willingness to use telemedicine technologies (Figure 2). These findings confirm earlier studies that highlight the importance of a stable technical infrastructure for the success of telemedicine interventions [27,37,38].

These technical difficulties increase the perceived effort and thus reduce the willingness to use the technology (effort expectancy). In addition, concerns about technical difficulties reflect participants’ expectations that the technologies should be easy to use (perceived ease of use).

In addition to technical hurdles, concerns regarding data protection may influence acceptance (see Figure 3). These results are consistent with the findings of Meingast et al. (2006) [39] and other experts who point to the need for strict data protection measures [40,41,42].

Support from family and friends plays a crucial role in the adoption of new technologies (social influence). Data protection concerns and mistrust of technology increase the perceived risk and reduce acceptance (perceived risk).

Social support also plays a crucial role in the introduction of new technologies. A lack of support from family or friends can reduce the acceptance of telemedicine solutions, while a supportive social environment facilitates motivation and use [16,42]. In the survey, participants stated that support from their personal environment facilitates the use of telemedicine (Figure 4).

Despite these barriers, the results also identify key success factors. Intuitive usability, simple user interfaces, and the communication of positive user experiences can significantly increase acceptance. The survey clearly showed that the presentation of success stories significantly promotes willingness to use telemedicine (see Figure 5) [17].

Positive user experiences and success stories increase the perceived usefulness of the technologies. Practical training, demonstrations, and ongoing technical support are crucial for improving the skills of family caregivers and reducing reservations about the technology (facilitating conditions).

Practical training, demonstrations, and ongoing technical support are also considered crucial. Hotlines, online tutorials, and personal support can improve the skills of family caregivers and reduce reservations about the technology [42,43,44]. Overall, the results underscore that the combination of technical infrastructure, social support, and targeted training can significantly increase the acceptance and use of telemedicine technologies.

### 3.2. Role of AI-Powered Chatbots

Participants rated AI-powered chatbots as potentially helpful in providing technical, emotional, and informational support when using telemedicine technologies. 41% of respondents said they would use a chatbot to receive help using telemedicine (see Figure 6).

Participants assumed that chatbots could be useful for providing support with technical and emotional challenges (perceived usefulness). They also suggested that the hypothetical 24/7 availability of chatbots might reduce the perceived effort required to use such technologies (effort expectancy).

Respondents primarily expected the following functions: technical instructions, emotional support, and answers to general questions. 77% of participants rated emotional support as important (see Figure 7 and Figure 8). This suggests that chatbots not only simplify operation but could also break down psychological barriers by strengthening trust and self-efficacy [13,23,26].

Another aspect is improved communication with the medical team. Chatbots could provide regular updates and clear information to avoid misunderstandings and improve the quality of care [23,26]. Participants emphasized that data protection is essential when using chatbots (see Figure 9), which underscores the need for transparent data protection information and secure communication channels [45,46].

### 3.3. Functionality and Benefits of a Chatbot

The survey results show that an AI-powered chatbot should fulfill several core functions:Technical supportStep-by-step instructions: 97% of participants expect clear and understandable instructions for installing and using telemedicine devices (Figure 10).Troubleshooting: Technical support is considered essential to overcome expected difficulties with chatbots (Figure 11).

2.Emotional supportStress management: Chatbots could send reassuring messages, provide tips on stress management, and boost user confidence.Success stories: Positive experiences from other users promote motivation and trust in the technology.3.Information provisionGeneral questions: Chatbots should answer basic questions about telemedicine to reduce uncertainty (see Figure 7).4.User-friendlinessIntuitive user interface: 100% of respondents emphasized the importance of ease of use (Figure 12).Multilingualism and barrier-free communication: Improve access for different user groups, e.g., through voice control or text-to-speech [24].

The results show that these functions are not only desirable but necessary in order to overcome barriers to the use of telemedicine and improve the quality of care.

### 3.4. Data Protection and Security

Data protection and data security are key concerns for respondents. 78% of participants rated the protection of their personal and medical data as very important (Figure 13).

74% are in favor of data encryption and the use of secure communication channels (Figure 14).

Data protection concerns and the need for secure communication channels reflect perceived risk. Transparent data protection policies and security measures are crucial for gaining user trust (facilitating conditions).

Transparent privacy policies and two-factor authentication were also considered essential (Figure 15). Regular security updates and reviews are seen as necessary to build trust and promote acceptance of telemedicine [45,46].

Overall, concerns regarding data protection and technical stability are consistent across both telemedicine and AI-powered chatbots. Addressing these overlapping barriers with secure systems, intuitive interfaces, and supportive guidance is crucial for adoption.

### 3.5. Application Support and Training

Regular training, hotlines, and visual aids are considered essential to improve technical skills and facilitate use of telemedicine and chatbots (Figure 16, Figure 17 and Figure 18).

Training should also cover topics such as data protection and data security [39,40] in order to fully prepare users for safe use [41].

Combined with social support, targeted training and practical guidance are crucial for successful adoption across both telemedicine and chatbot technologies, highlighting the interplay of technical, social, and psychological dimensions.

### 3.6. Communication with the Medical Team

Efficient communication with medical staff is essential; participants highlighted difficulties in timely feedback and the need for emergency communication plans (Figure 19 and Figure 20).

Support from the medical team and improved communication increase acceptance of the technologies (social influence). Standardized communication protocols and emergency communication plans are crucial for improving the quality of care (facilitating conditions).

### 3.7. Summary of Results

The surveys show that the successful integration of telemedicine and AI-powered chatbots depends on several factors:Technical support: Step-by-step instructions and troubleshooting are essential.Emotional support: Stress management and motivation through chatbots increase acceptance.Provision of information: Clear answers and data protection information reduce uncertainty.User-friendliness: Intuitive operation and multilingualism facilitate use.Data protection and security: Transparent guidelines, encryption, and two-factor authentication build trust.Training and support: Continuous training and technical support improve skills and acceptance.Communication with the medical team: Standardized protocols and emergency communication plans increase the quality of care.

The results confirm the findings of existing literature [13,15,16,45,46] and show that AI-supported chatbots could offer valuable support to family caregivers [23,26].

The assignment of the results to the constructs of the Technology Acceptance Model (TAM) and the Unified Theory of Acceptance and Use of Technology (UTAUT) illustrates how various factors influence the acceptance and use of telemedicine technologies and AI-supported chatbots (Table 7). The results show that both technical and psychosocial aspects must be taken into account to ensure the successful implementation and use of these technologies.

## 4. Discussion

This study builds on an earlier study on the integration of telemedicine into home care and expands on it with a follow-up study on the use of AI-supported chatbots. While the first study identified structural, organizational, and technical barriers to the introduction of telemedicine systems in family households, the second survey focuses on the communicative and interactive dimension of digital support. Combining both data sets provides a more comprehensive picture of the digital transformation process in home care.

### 4.1. Technical Support and User-Friendliness

Technical difficulties, such as connection problems, are a common barrier for both telemedicine and chatbots. Chatbots may help overcome these hurdles through clear step-by-step instructions, thus enhancing usability [13,23,26,31,32,33,34,35,47].

### 4.2. Emotional Support and Stress Management

Chatbots are perceived as potentially providing not only functional but also emotional support, for example, by offering reassurance, stress management tips, or motivational success stories, which participants believe could enhance caregiver confidence and self-efficacy [13,23,26,31,32,33,34,35,48].

### 4.3. Information Provision and Data Protection

Data protection remains a key barrier for both telemedicine and chatbots. Transparent privacy information and secure communication channels are essential to build user trust [45,46,47].

### 4.4. Communication with the Medical Team

Chatbots could improve communication with the medical team by providing regular updates and facilitating emergency contacts, thereby supporting timely information exchange and high quality of care [22,23,25,32,48].

### 4.5. Comparison of the Results of Both Surveys

Comparing both surveys shows that while telemedicine adoption depends largely on technical infrastructure, chatbot acceptance additionally relies on comprehensibility, empathy, and trust. This illustrates the shift from technology as a tool to technology as a social interaction partner [23,26,31,32,33,34,35,47].

### 4.6. Innovation and Added Value of the Chatbot Component

AI-powered chatbots enable continuous, low-threshold interaction that combines informative and emotional support. However, family caregivers emphasize that chatbots should complement human contact, not replace it [13,23,26,31,32,33,34,35]. This ambivalence can be understood as an expression of dual expectations: family caregivers want digital support, but at the same time demand emotional authenticity and personal resonance [13,23,26,31,32,33,34,35,48].

Compared to telemedicine, which primarily strengthens the structural connection between home care and professional care [27,37,38], the use of chatbots could represent a person-centered advance. The technology addresses not only organizational but also psychosocial challenges [13,23,26,31,32,33,34,35]. It can help reduce cognitive and emotional stress by acting as a communicative link between people and the system It is particularly noteworthy that caregiving relatives hypothetically assume that chatbots are more likely to be perceived as companions in everyday care than as a primary source of medical information [23,26,31,32,33,34,35]. This shift in perspective points to a new phase of digitalization in the context of caregiving, one that is more focused on interaction and relationship building [13,23,26,31,32,33,34,35].

### 4.7. Practical Implications

The practical implementation of AI chatbots in home care should be participatory, including training for family caregivers, and ensure transparency, data protection, and emotional security. Trust in the functionality of the system is essential to realize the full potential of digital support [13,23,26,31,32,33,34,35,45,46,47].

### 4.8. Scientific Implications

Scientifically, the integration of AI chatbots expands the concept of digital caregiving competence to include emotional and communicative dimensions. Future research should examine the long-term effects on the well-being of family caregivers and identify design principles that promote trust, empathy, and transparency [13,23,26,31,32,33,34,35,48].

### 4.9. Summary of the Discussion

Overall, this study shows that digital support for family caregivers is undergoing a phase of qualitative transformation. The transition from purely functional telemedicine applications to AI-based, interactive systems marks a paradigm shift: from technical networking to communicative cooperation. The study thus makes an innovative contribution to the further development of digital care models in the home context and at the same time provides practice-relevant insights for the design of sustainable care support [13,23,26,31,32,33,34,35,48]. However, it is important to emphasize that the results are exploratory and descriptive and should serve as a basis for further research.

### 4.10. Critical Reflection and Limitations

Despite the insights gained, the study has several limitations. First, it is a cross-sectional online survey that only captures snapshots of attitudes and perceptions. Long-term studies would be necessary to examine how acceptance and user experiences with AI-based chatbots in everyday care develop over time.

Second, the sample is not representative of all family caregivers in Germany. People with a higher digital affinity may be overrepresented, which could lead to a certain bias toward technology-friendly attitudes. Future studies should therefore specifically include more heterogeneous groups, in particular older, less digitally experienced, or socially disadvantaged family caregivers.

Third, the assessments of the use of chatbots are based on hypothetical scenarios, as many respondents have not yet gained any practical experience with AI systems. Accordingly, the results should be interpreted as indicators of expectations and perceptions, not as evidence of actual usage behavior. Experimental or qualitative follow-up studies could provide valuable additions here.

Fourth, participants were selected based on their participation in a caregiving course, which may lead to selection bias. Participants may be more motivated and digitally savvy than the general population of family caregivers, which limits the generalizability of the results.

Finally, the study was limited to the perspective of family caregivers. The inclusion of professional caregivers, consultants, and developers could provide a more comprehensive understanding of how AI-supported systems can be integrated into existing care structures in the future.

In addition to these points, there are further limitations:Sample size and representativeness: The first survey had only 23 participants, while the second survey had 39 participants. These relatively small sample sizes limit the generalizability of the results. Future studies should include larger and more diverse samples to obtain more representative results.Time lag: The surveys were conducted at different times (January vs. September 2025). This time lag may have influenced learning effects and changes in participants’ experiences and attitudes. It would be useful to conduct future studies closer together in time to minimize such effects.Geographical and cultural limitations: Both surveys were conducted with national participants, which limits the transferability of the results to international contexts. Cultural differences in the acceptance and use of telemedicine and chatbots could influence the results. Future research should include international samples to account for cultural differences.Self-reported data: The data is based on self-reported information from participants, which may lead to bias due to social desirability or memory effects. Supplementary objective data, such as usage statistics or observations, could increase the validity of the results.Technological development: Rapid technological development in the field of AI and telemedicine could cause the results to quickly become outdated. It is important to conduct ongoing research to keep pace with the latest developments and understand their impact on home care.

Despite these limitations, the study offers important starting points for research and practice. It makes it clear that AI technologies in the care context must be understood primarily as communicative and social innovations rather than technical ones. The resulting change in perspective can make a decisive contribution to shaping digital transformation in care in a human-centered, participatory, and ethically responsible manner. The results underscore the need for further research to better understand and optimize the acceptance, use, and long-term impact of these technologies [49].

## 5. Conclusions

This study shows that the use of AI-powered chatbots in home care has the potential to significantly expand the current development of digital support systems. Building on the results of the previous telemedicine study, it is clear that the perspective of family caregivers is shifting from purely functional expectations to an interactive, dialogue-oriented understanding of digital support [13,23,26,31,32,33,34,35].

AI-powered chatbots could provide not only technical support, but also emotional and social relief. They could overcome technical barriers through clear instructions and problem solving and increase user-friendliness. In addition, they could offer emotional support through reassuring messages, stress management tips, and sharing success stories, which would boost users’ self-confidence and self-efficacy [13,23,26,31,32,33,34,35].

Privacy concerns were identified as a key barrier in both surveys [39,40,41]. Chatbots could strengthen user trust through transparent privacy information and secure communication channels [40,41]. The results show that chatbots could improve communication with the medical team by providing regular updates and standardized communication protocols [22,23,25,32].

In practice, this leads to the recommendation that AI systems should be specifically integrated into counseling and training programs for family caregivers. Politically and institutionally, framework conditions are needed that promote innovation without neglecting ethical and data protection principles. From a scientific perspective, the results underscore the need to expand the concept of “digital care competence” to include communicative and emotional dimensions. Future studies should investigate the extent to which such systems can strengthen the self-efficacy and well-being of family caregivers in the long term [13,23,26,31,32,33,34,35].

It is important to emphasize that the results of this study are exploratory and descriptive. The relatively small sample size and the possible overlap of participants between the two studies limit the generalizability of the results. In addition, the assessments of chatbot use are based on hypothetical scenarios, as many participants have not yet gained practical experience with these systems.

Overall, this study shows that digital support for family caregivers is undergoing a phase of qualitative transformation. The transition from purely functional telemedicine applications to AI-based, interactive systems marks a paradigm shift: from technical networking to communicative collaboration. The results underscore the need for further research to better understand and optimize the acceptance, use, and long-term impact of these technologies. Through the continuous development and adaptation of these technologies, the quality of home care can be improved and the burden on family caregivers can be reduced in the long term [13,23,26,31,32,33,34,35].

## Figures and Tables

**Figure 1 healthcare-13-03159-f001:**
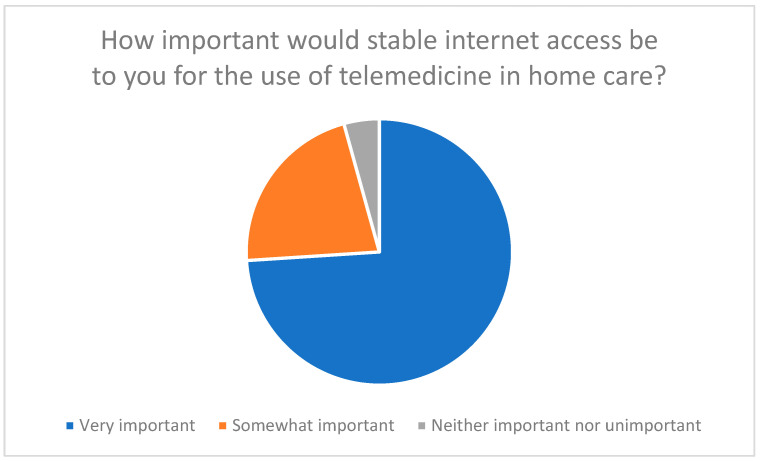
Assessment of the importance of stable internet access for the use of telemedicine in home care.

**Figure 2 healthcare-13-03159-f002:**
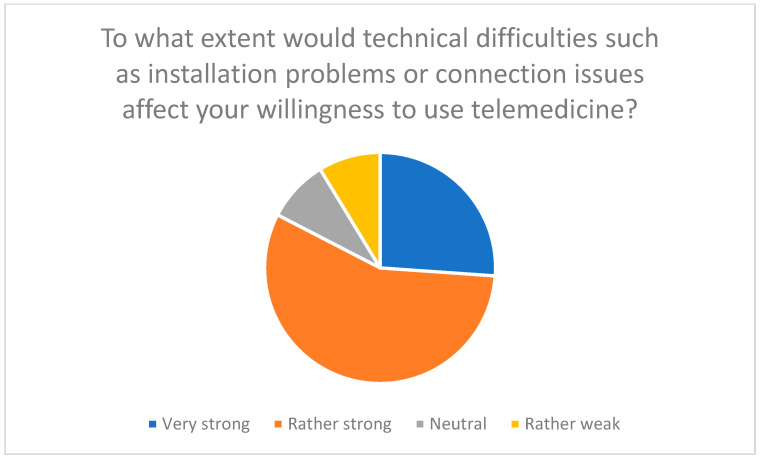
Influence of technical difficulties such as installation or connection problems on willingness to use telemedicine.

**Figure 3 healthcare-13-03159-f003:**
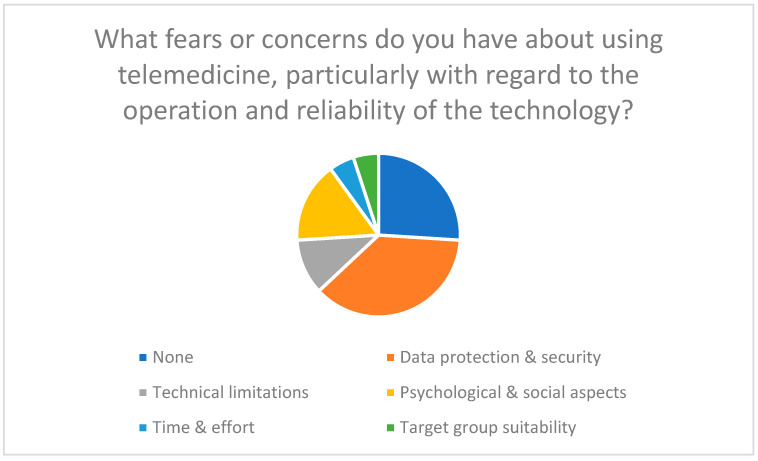
Fears and concerns regarding the use of telemedicine, particularly in relation to user-friendliness and reliability of the technology.

**Figure 4 healthcare-13-03159-f004:**
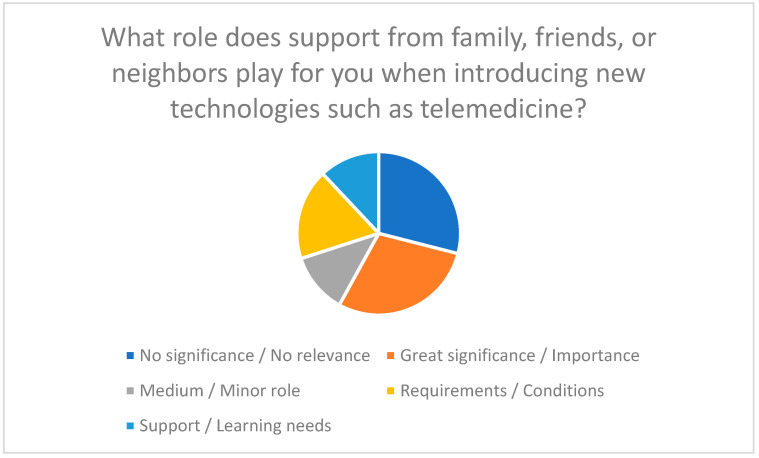
Importance of support from family, friends, or neighbors when introducing new technologies such as telemedicine.

**Figure 5 healthcare-13-03159-f005:**
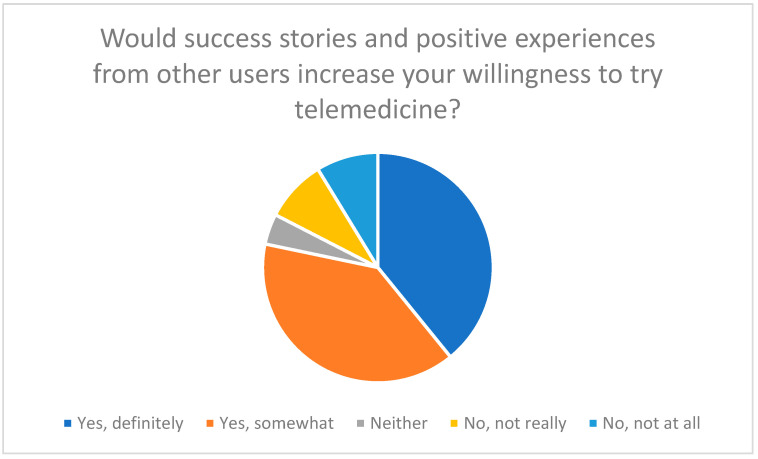
Influence of success stories and positive experiences of other users on willingness to try telemedicine.

**Figure 6 healthcare-13-03159-f006:**
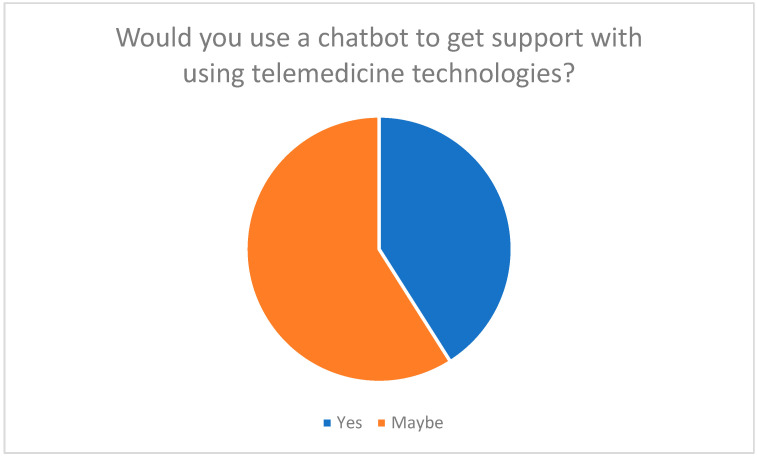
Willingness to use a chatbot to assist with the application of telemedicine technologies.

**Figure 7 healthcare-13-03159-f007:**
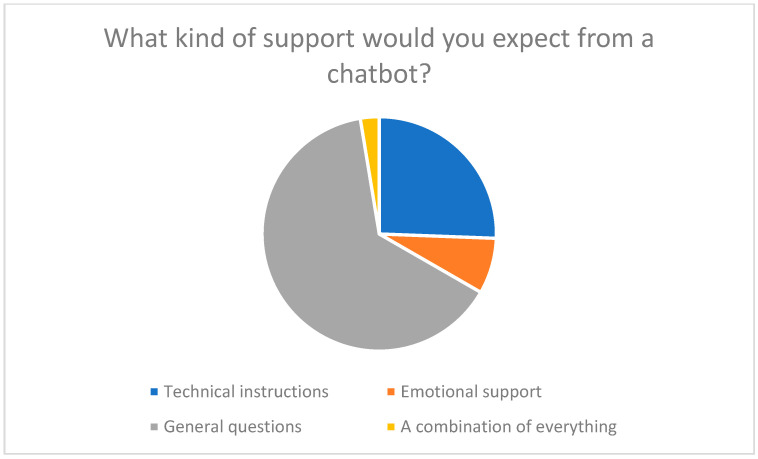
What kind of support would you expect from a chatbot? (Technical instructions, Emotional support, General questions, Other).

**Figure 8 healthcare-13-03159-f008:**
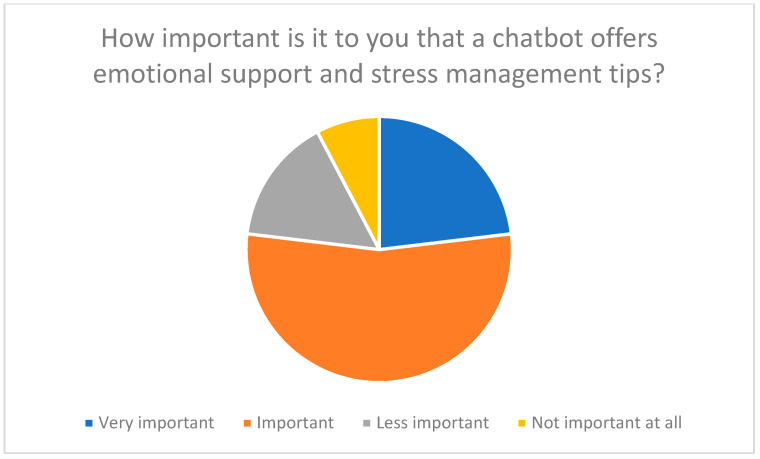
The importance of emotional support from a chatbot for family caregivers.

**Figure 9 healthcare-13-03159-f009:**
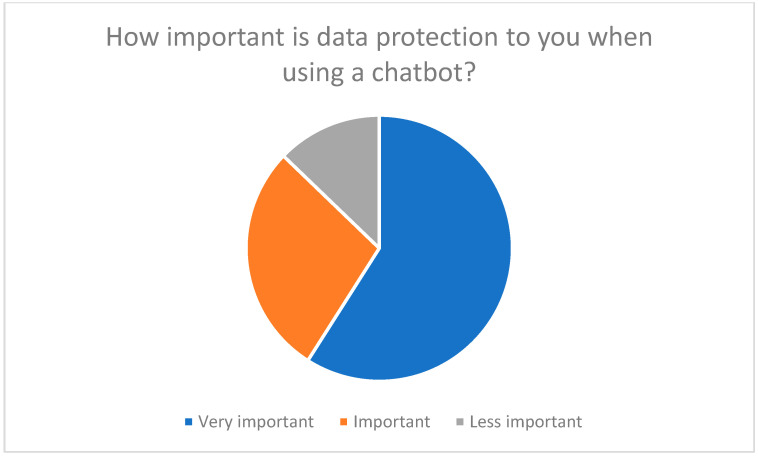
Concerns about data protection when using chatbots.

**Figure 10 healthcare-13-03159-f010:**
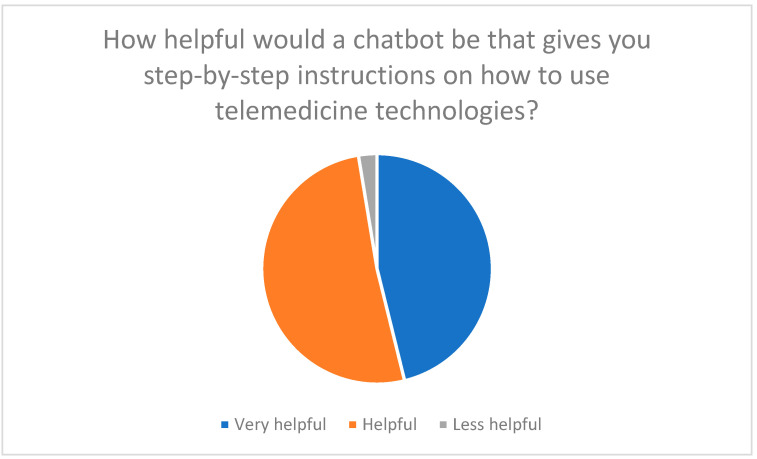
Expectation of step-by-step instructions from a chatbot.

**Figure 11 healthcare-13-03159-f011:**
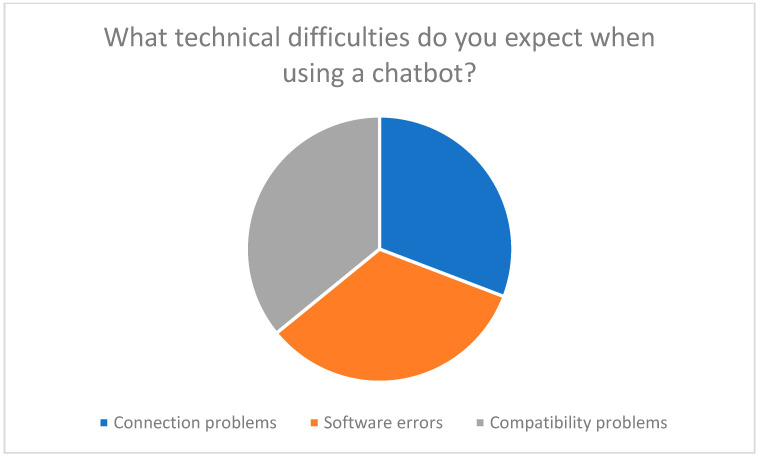
Technical difficulties when using a chatbot.

**Figure 12 healthcare-13-03159-f012:**
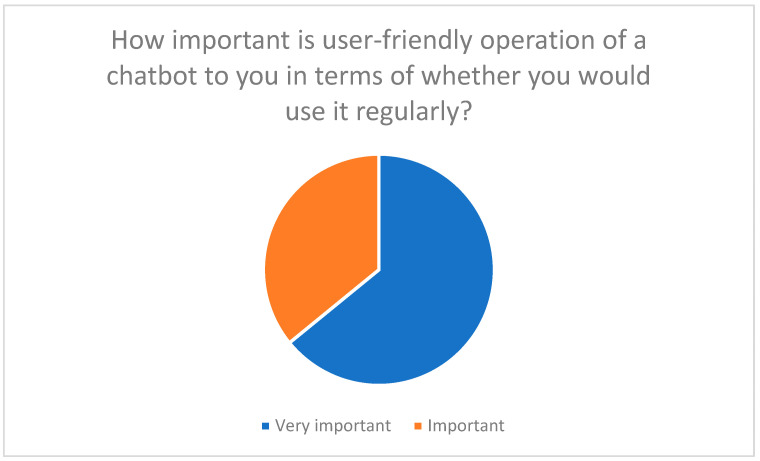
The importance of an intuitive user interface for the acceptance of chatbots.

**Figure 13 healthcare-13-03159-f013:**
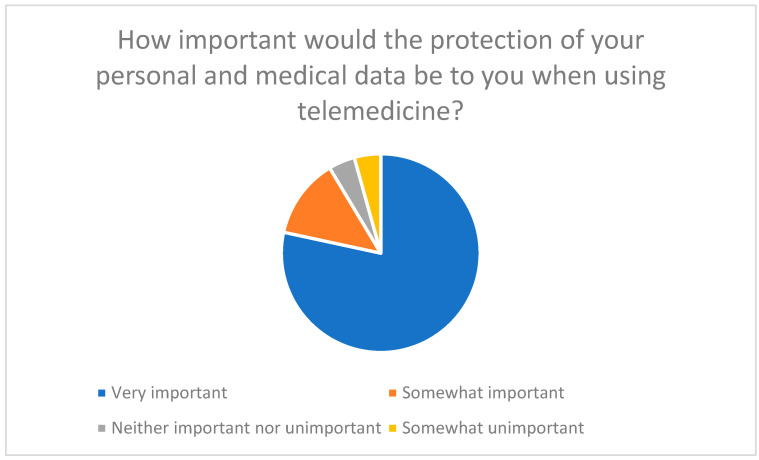
The importance of protecting personal and medical data when using telemedicine.

**Figure 14 healthcare-13-03159-f014:**
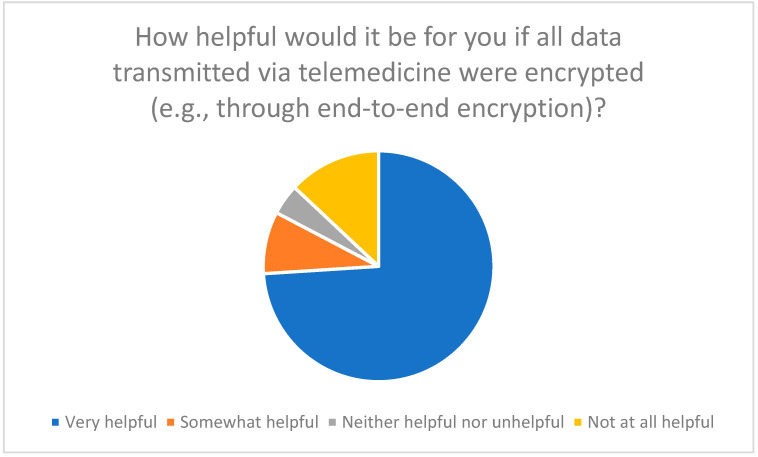
Assessment of the usefulness of data encryption (e.g., end-to-end encryption) when using telemedicine.

**Figure 15 healthcare-13-03159-f015:**
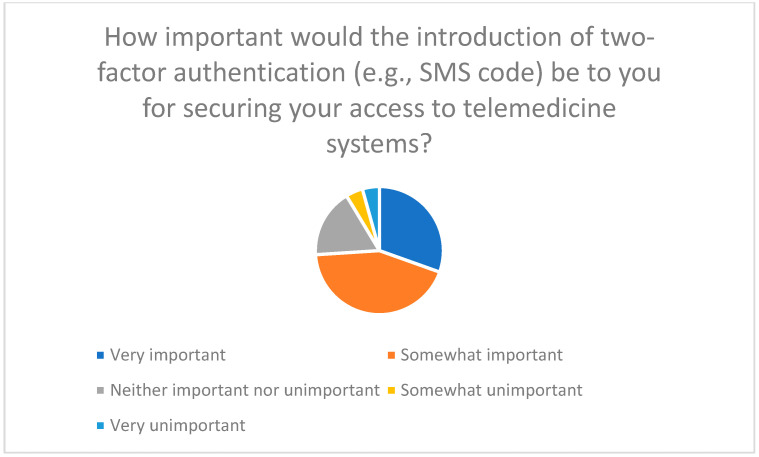
Importance of introducing two-factor authentication to secure access to telemedicine systems.

**Figure 16 healthcare-13-03159-f016:**
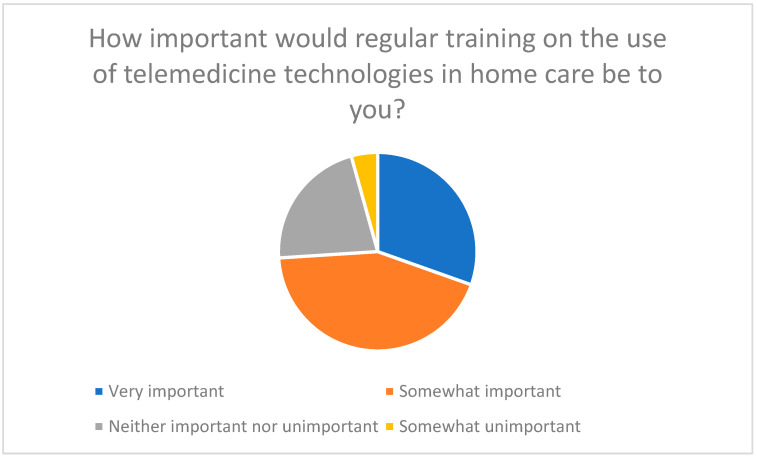
The importance of regular training on the use of telemedicine technologies in home care.

**Figure 17 healthcare-13-03159-f017:**
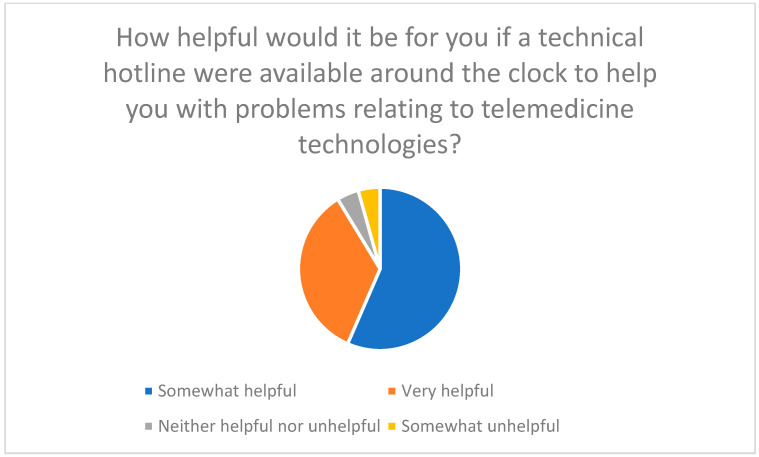
Usefulness of a 24/7 technical hotline for telemedicine issues.

**Figure 18 healthcare-13-03159-f018:**
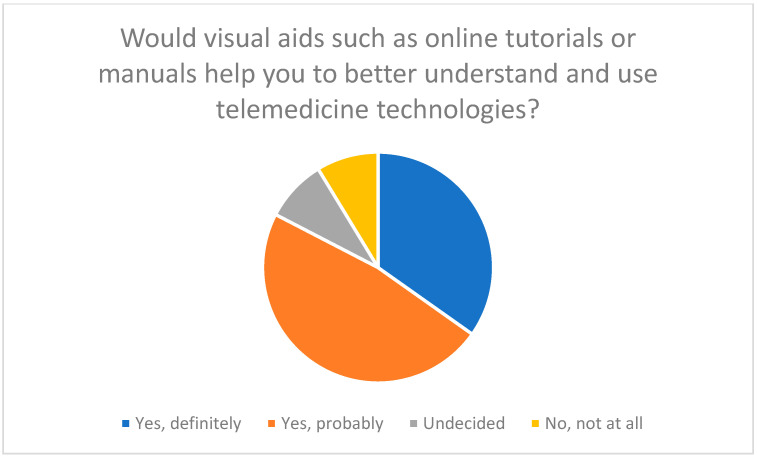
Importance of visual aids to support the use of telemedicine technologies.

**Figure 19 healthcare-13-03159-f019:**
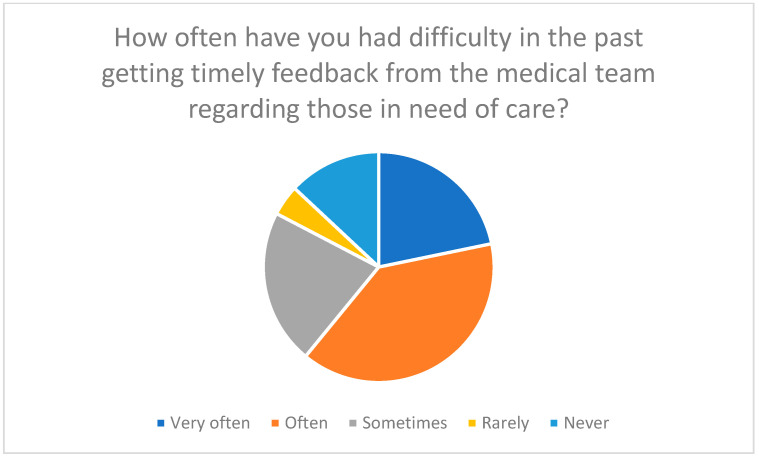
Frequency of difficulties in receiving timely feedback from the medical team.

**Figure 20 healthcare-13-03159-f020:**
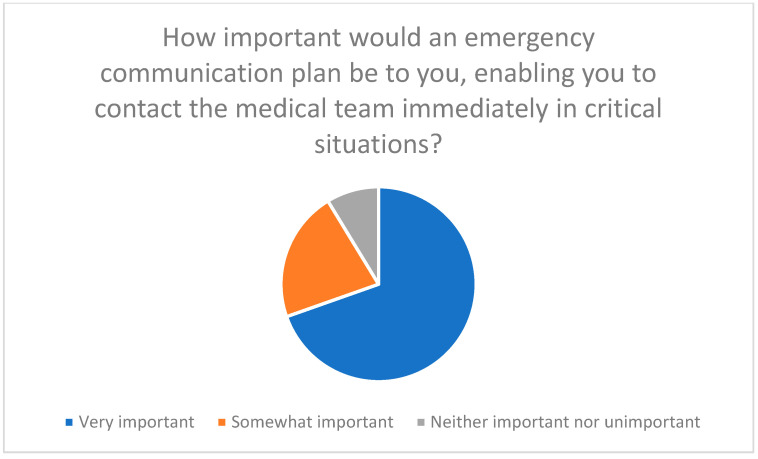
Importance of an emergency communication plan that enables immediate contact with the medical team in critical situations.

**Table 1 healthcare-13-03159-t001:** Demographic characteristics and caregiving experience of survey participants (study 1).

Variable	Category	*n*
Age	Under 30	8
	30–39	4
	40–49	5
	50–59	4
	60–69	2
Gender	Male	4
	Female	17
	Diverse	2
Educational attainment	Secondary school	4
	High school diploma	4
	Vocational training	9
	University degree	5
	Other	1
Care experience	<6 months	8
	6 months–1 year	3
	1–3 years	3
	>3 years	9
Care expenditure/week	<10 h	5
	10–20 h	12
	21–30 h	4
	>30 h	2

**Table 2 healthcare-13-03159-t002:** Demographic characteristics and caregiving experience of survey participants (study 2).

Variable	Category	*n*
Age	Under 30	4
	30–39	6
	40–49	16
	50–59	9
	60–69	3
	70+	1
Gender	Male	6
	Female	33
Educational attainment	Secondary school	8
	High school diploma	5
	Vocational training	11
	University degree	15

**Table 3 healthcare-13-03159-t003:** List of sample items in Study 1.

Main Topic	Sample Item	i-CVI
General experiences	Q1: Have you already had experience with the use of telemedicine?	0.88
	Q3: What was your overall experience with using telemedicine?	0.89
Acceptance and trust	Q7: How important would intuitive and easy-to-understand operation of telemedicine be to you?	0.87
	Q16: How important would the protection of your personal and medical data be to you when using telemedicine?	0.87
Perceived barrier	Q11: What technical barriers could prevent you from using telemedicine?	0.89
	Q12: What fears or concerns do you have about using telemedicine?	0.88
Support and training needs	Q27: How important would regular training on the use of telemedicine technologies in home care be to you?	0.88
	Q28: How helpful would it be for you if a technical hotline were available around the clock?	0.87

**Table 4 healthcare-13-03159-t004:** List of sample items in Study 2.

Main Topic	Sample Item	i-CVI
Communication and interaction	Q1: How helpful would an AI-powered chatbot be for technical support when using telemedicine?	0.92
	Q3: How helpful would an AI-powered chatbot be for providing general information about telemedicine?	0.91
Trust in chatbots	Q6: How much do you trust the reliability of an AI-powered chatbot?	0.91
	Q7: How much do you trust the security of an AI-powered chatbot?	0.90
Emotional support	Q8: How helpful would it be for you if an AI-powered chatbot could provide stress management tips?	0.92
	Q9: How helpful would it be for you if an AI-powered chatbot could share success stories from other users?	0.91
Data protection and security	Q10: How important is the protection of your personal and medical data when using an AI-powered chatbot?	0.90
	Q11: How much would concerns about data protection affect your willingness to use an AI-powered chatbot?	0.93

**Table 5 healthcare-13-03159-t005:** Subscale assignment including Cronbach’s alpha, i-CVI, and theoretical reference TAM and UTAUT (study 1).

Subscale	Items	Cronbach’s α	i-CVI Per Item	Theoretical Reference (Model Component)
Context/experience	1, 2, 3, 4, 48, 49, 50, 51, 52	α = 0.82	0.83–1.0	Perceived Usefulness
Acceptance/trust	7, 8, 10, 12, 14, 15, 16, 17, 18, 19, 20, 21, 22, 23, 24, 25, 26	α = 0.87	0.83–1.0	Perceived Usefulness
Barriers	5, 6, 9, 11, 30, 34	α = 0.75	0.83–1.0	Effort Expectancy/Perceived Ease of Use
Training/support	13, 27, 28, 29, 31, 32, 33, 35, 36	α = 0.80	0.83–1.0	Facilitating Conditions
Communication/collaboration	37, 38, 39, 40, 41, 42, 43, 44, 45, 46, 47	α = 0.85	0.83–1.0	Social Influence

**Table 6 healthcare-13-03159-t006:** Subscale assignment including Cronbach’s alpha, i-CVI, and theoretical reference TAM and UTAUT (study 2).

Subscale	Items	Cronbach’s α	i-CVI Per Item	Theoretical Reference (Model Component)
Communication/Interaction	1, 2, 3, 4, 5	α = 0.82	0.83–1.0	Social Influence/Effort Expectancy
Trust in chatbots	6, 7	α = 0.81	0.83–1.0	Perceived Usefulness
Emotional support	8, 9	α = 0.79	0.83–1.0	Social Influence
Data protection/Security	10–15	α = 0.83	0.83–1.0	Perceived Risk/Facilitating Conditions

**Table 7 healthcare-13-03159-t007:** Assignment of result areas to the TAM/UTAUT construct.

Result Area	TAM/UTAUT Construct	Description
Acceptance and use of digital health technologies	Perceived Usefulness, Effort Expectancy	Stable internet access and suitable devices increase perceived usefulness (Perceived Usefulness) and reduce effort (Effort Expectancy).
Technical barriers	Effort Expectancy, Perceived Ease of Use	Technical difficulties increase perceived effort (Effort Expectancy) and reduce user-friendliness (Perceived Ease of Use).
Psychological barriers and social support	Social Influence, Perceived Risk	Support from family and friends (Social Influence) as well as privacy concerns (Perceived Risk) influence acceptance.
Success factors	Perceived Usefulness, Social Influence	Positive user experiences and success stories increase perceived benefits (Perceived Usefulness) and are reinforced by social support (Social Influence).
Role of AI-powered chatbots	Perceived Usefulness, Effort Expectancy	Chatbots could offer support with technical and emotional challenges (perceived benefit) and reduce effort (effort expectation).
Privacy concerns	Perceived Risk, Facilitating Conditions	Privacy concerns increase the perceived risk (Perceived Risk) and require measures to facilitate use (Facilitating Conditions).
Training and support needs	Facilitating Conditions	Training and technical support facilitate use and increase acceptance (Facilitating Conditions).
Communication with the medical team	Social Influence, Facilitating Conditions	Support from the medical team (Social Influence) and improved communication (Facilitating Conditions) increase acceptance.

## Data Availability

The data are contained within the article.

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
