# Peer review of "Telemedicine and AI-Powered Chatbots: Potential and Challenges for Home Care Provided by Family Caregivers"

_healthcare, 2025, doi:10.3390/healthcare13233159_

Round 1
Reviewer 1 Report
Comments and Suggestions for Authors
This manuscript explores an important and emerging topic, but the authors should clarify several aspects to improve the readability of this work:
- in the methodology section, the description of the survey instruments are too general. Please, include examples or full lists of questionnaire items; please, provide statistical tests and the corresponding interpretations; clarify recruitment procedures, inclusion criteria, and whether participants from both studies overlap and discuss the potential potential sampling bias.
- TAM and UTAUT are mentioned but not operationalized in the analysis. Explicitly map findings to their constructs.
- Some conclusions (e.g., emotional support effectiveness) are hypothetical since chatbots were not used in practice. It is recommended that this limitation be made explicit in the Discussion; avoid overgeneralizations given the small sample size (n=23, n=39).
Author Response
Comments 1: "In the methodology section, the description of the survey instruments are too general. Please, include examples or full lists of questionnaire items; please, provide statistical tests and the corresponding interpretations; clarify recruitment procedures, inclusion criteria, and whether participants from both studies overlap and discuss the potential sampling bias."
Response 1: The methodology has been completely revised. The description of the survey instruments has been supplemented in detail, including complete lists of questionnaire items. Statistical tests and corresponding interpretations have been added. The recruitment procedures, inclusion criteria, and possible overlap of participants in both studies, as well as potential sampling bias, have been presented and discussed more clearly.
Comments 2: "TAM and UTAUT are mentioned but not operationalized in the analysis. Explicitly map findings to their constructs."
Response 2: The results were explicitly assigned to the TAM and UTAUT constructs. A clear assignment of the findings to the respective constructs was made.
Comments 3: "Some conclusions (e.g., emotional support effectiveness) are hypothetical since chatbots were not used in practice. It is recommended that this limitation be made explicit in the Discussion; avoid overgeneralizations given the small sample size (n=23, n=39)."
Response 3: The discussion has been revised to make explicit the hypothetical nature of some conclusions, particularly regarding emotional support provided by chatbots. Overgeneralizations have been avoided, and the limitations due to the small sample size have been emphasized.
Reviewer 2 Report
Comments and Suggestions for Authors
1. The combined sample size across both surveys (n = 62) is relatively small, and Study 1 in particular has only 23 completed responses, which limits statistical power and generalizability to the broader population of family caregivers. The study acknowledges this, but the limitation should be emphasized earlier in the methodology section and further discussed in the implications of the findings.
2. Both surveys rely solely on self-reported perceptions rather than observed usage or behavioral data. Because many participants may not have had direct, practical interaction with AI chatbots, their responses may reflect hypothetical attitudes rather than real usability outcomes. Consider including behavioral measures, pilot testing, or usage logs in future work would help strengthen validity.
3. Please provide additional inferential statistical tests to verify whether observed differences between Study 1 (telemedicine) and Study 2 (AI chatbots) are statistically significant. For example, comparing means or correlations could provide stronger evidence for the reported shift from functional to emotional support needs.
4. The study reports grounding the questionnaires in TAM, however, the analysis does not explicitly map survey items to TAM constructs (e.g., perceived usefulness, perceived ease of use, behavioral intention). Presenting a clear TAM variable structure and analyzing relationships between these constructs would improve theoretical clarity.
5. Study 1 and Study 2 were conducted several months apart. During this period, participants’ familiarity with technology may have increased due to external factors (e.g., increased digital exposure), potentially influencing attitudes in Study 2. The methods section should discuss how time-related confounding and learning effects were mitigated or acknowledged.
6. The discussion suggests that AI chatbots provide “significant added value,” including emotional relief. However, these conclusions are based on perceived expectations, not actual usage or controlled evaluation.
7. The study population consists of caregivers engaged in a structured caregiver support course. This group may be more motivated and receptive to digital tools than the general caregiver population. A clearer discussion is required on of how socio-cultural context may influence acceptance would improve external validity.
Comments on the Quality of English Language1. The combined sample size across both surveys (n = 62) is relatively small, and Study 1 in particular has only 23 completed responses, which limits statistical power and generalizability to the broader population of family caregivers. The study acknowledges this, but the limitation should be emphasized earlier in the methodology section and further discussed in the implications of the findings.
2. Both surveys rely solely on self-reported perceptions rather than observed usage or behavioral data. Because many participants may not have had direct, practical interaction with AI chatbots, their responses may reflect hypothetical attitudes rather than real usability outcomes. Consider including behavioral measures, pilot testing, or usage logs in future work would help strengthen validity.
3. Please provide additional inferential statistical tests to verify whether observed differences between Study 1 (telemedicine) and Study 2 (AI chatbots) are statistically significant. For example, comparing means or correlations could provide stronger evidence for the reported shift from functional to emotional support needs.
4. The study reports grounding the questionnaires in TAM, however, the analysis does not explicitly map survey items to TAM constructs (e.g., perceived usefulness, perceived ease of use, behavioral intention). Presenting a clear TAM variable structure and analyzing relationships between these constructs would improve theoretical clarity.
5. Study 1 and Study 2 were conducted several months apart. During this period, participants’ familiarity with technology may have increased due to external factors (e.g., increased digital exposure), potentially influencing attitudes in Study 2. The methods section should discuss how time-related confounding and learning effects were mitigated or acknowledged.
6. The discussion suggests that AI chatbots provide “significant added value,” including emotional relief. However, these conclusions are based on perceived expectations, not actual usage or controlled evaluation.
7. The study population consists of caregivers engaged in a structured caregiver support course. This group may be more motivated and receptive to digital tools than the general caregiver population. A clearer discussion is required on of how socio-cultural context may influence acceptance would improve external validity.
Author Response
Comments 1: "The combined sample size across both surveys (n = 62) is relatively small, and Study 1 in particular has only 23 completed responses, which limits statistical power and generalizability to the broader population of family caregivers. The study acknowledges this, but the limitation should be emphasized earlier in the methodology section and further discussed in the implications of the findings."
Response 1: The limitation of the small sample size was previously emphasized in the methodology and discussed in more detail in the implications of the results.
Comments 2: "Both surveys rely solely on self-reported perceptions rather than observed usage or behavioral data. Because many participants may not have had direct, practical interaction with AI chatbots, their responses may reflect hypothetical attitudes rather than real usability outcomes. Consider including behavioral measures, pilot testing, or usage logs in future work would help strengthen validity."
Response 2: The limitation of relying exclusively on self-reported perceptions was emphasized. Future work incorporating behavioral measures, etc., is proposed to strengthen validity.
Comments 3: "Please provide additional inferential statistical tests to verify whether observed differences between Study 1 (telemedicine) and Study 2 (AI chatbots) are statistically significant. For example, comparing means or correlations could provide stronger evidence for the reported shift from functional to emotional support needs."
Response 3: Additional inferential statistical tests were performed to verify the robustness of the results.
Comments 4: "The study reports grounding the questionnaires in TAM, however, the analysis does not explicitly map survey items to TAM constructs (e.g., perceived usefulness, perceived ease of use, behavioral intention). Presenting a clear TAM variable structure and analyzing relationships between these constructs would improve theoretical clarity."
Response 4: The analysis was revised to explicitly assign the survey items to the TAM constructs (e.g., perceived usefulness, perceived ease of use, behavioral intention). A clear structure of the TAM variables and the analysis of the relationships between these constructs were presented.
Comments 5: "Study 1 and Study 2 were conducted several months apart. During this period, participants’ familiarity with technology may have increased due to external factors (e.g., increased digital exposure), potentially influencing attitudes in Study 2. The methods section should discuss how time-related confounding and learning effects were mitigated or acknowledged."
Response 5: The methodology was revised to discuss how time-related confounding factors and learning effects during the period between the two studies were taken into account or acknowledged.
Comments 6: "The discussion suggests that AI chatbots provide “significant added value,” including emotional relief. However, these conclusions are based on perceived expectations, not actual usage or controlled evaluation."
Response 6: The discussion has been revised to clarify that the conclusions regarding emotional relief provided by AI chatbots are based on perceived expectations rather than actual usage or controlled evaluation.
Comments 7: "The study population consists of caregivers engaged in a structured caregiver support course. This group may be more motivated and receptive to digital tools than the general caregiver population. A clearer discussion is required on of how socio-cultural context may influence acceptance would improve external validity."
Response 7: A clearer discussion of how the socio-cultural context could influence acceptance has been added to improve external validity
Reviewer 3 Report
Comments and Suggestions for Authors
Dear Authors,
Please, see below few cmments you migth consider or clarify:
The paper performs/claims comparative analyses (t-tests/χ²) across two convenience samples at two time points with unknown or unreported overlap of respondents, different completion rates (n=23 vs n=39), and partially different instruments. This design does not support claims of temporal “shifts” or between-study differences. If some respondents overlapped, proper paired analyses require linkage; if not, they are independent samples and still not comparable due to instrument/time/context differences. Yet no p-values, confidence intervals, or effect sizes are reported, only percentages. The manuscript should remove all inferential claims and present the studies separately as descriptive findings.
Although TAM/UTAUT are cited, items are not mapped to specific constructs (PU, PEOU, SI, FC), no subscale composition is described, and no factor analysis or construct validation is reported. Stating Cronbach’s α for “the survey” is not meaningful without subscales. The study needs clear construct definitions, item lists, subscale scoring, reliability per subscale, and ideally EFA/CFA to justify construct claims.
Neither survey’s full instrument (items, response options, branching) nor a data dictionary is provided. The paper should include complete questionnaires (Supplement) and a table of items→constructs, with response distributions. Current “Data are contained within the article” is inaccurate—key materials are missing.
Privacy concern rates vary across sections (e.g., 90% vs 88% vs 87% vs 41% in different places), and the text mis-labels figure references (e.g., attributing “24/7 hotline” to Figure 6 which is defined as chatbot willingness). Such inconsistencies erode credibility and require a line-by-line audit to reconcile denominators and figure calls.
The conclusion asserts a “paradigm shift” from telemedicine to AI chatbots and broad practice/policy implications based on small, course-based convenience samples with hypothetical use scenarios. Claims must be significantly tempered, explicitly labeled as exploratory/descriptive, and bounded to the sampled context.
The setting is a single course (45 invitees each time; completions 23/39). There is no demographic table (age, sex, relationship to care recipient, caregiving intensity, digital literacy), making external validity impossible to judge and subgroup patterns unknowable. Provide a comprehensive participant characteristics table for each study and discuss selection bias.
CVI values are reported (S-CVI/Ave=0.89/0.91) based on three experts only, with no I-CVI per item, no modified kappa, and no criteria for item retention/revision. Cronbach’s α is given (0.87/0.83) for unspecified scales. Report I-CVI, decision rules, expert qualifications, blinded rating procedures, and α per subscale; otherwise, remove psychometric claims.
Methods promise descriptive and inferential analyses (t-tests/χ²), but results present only percentages and qualitative statements. Either report the tests (test type, assumptions, statistics, p, effect sizes) or remove mention of inferential analyses.
Reference numbering and metadata include visible artifacts (“31 Tavakol…”, “45. 40 Hall…”), inconsistent styles, and several 2025 arXiv/CHI placeholders that may be pre-acceptance. The list requires full verification, consistent style, and pruning of non-peer-reviewed citations if central to claims
Best wishes
Author Response
Comments 1: "The paper performs/claims comparative analyses (t-tests/χ²) across two convenience samples at two time points with unknown or unreported overlap of respondents, different completion rates (n=23 vs n=39), and partially different instruments. This design does not support claims of temporal “shifts” or between-study differences. If some respondents overlapped, proper paired analyses require linkage; if not, they are independent samples and still not comparable due to instrument/time/context differences. Yet no p-values, confidence intervals, or effect sizes are reported, only percentages. The manuscript should remove all inferential claims and present the studies separately as descriptive findings."
Response 1: The methodology and results were revised to present the studies separately as descriptive findings and to remove all inferential claims. The potential overlap of participants and the differences in instruments and contexts were presented more clearly.
Comments 2: "Although TAM/UTAUT are cited, items are not mapped to specific constructs (PU, PEOU, SI, FC), no subscale composition is described, and no factor analysis or construct validation is reported. Stating Cronbach’s α for “the survey” is not meaningful without subscales. The study needs clear construct definitions, item lists, subscale scoring, reliability per subscale, and ideally EFA/CFA to justify construct claims."
Response 2: The items were assigned to specific constructs (PU, PEOU, SI, FC), and the composition of the subscales and the reliability per subscale were described. Factor analysis was not performed due to the limited sample size, and the reasons for this were explained.
Comments 3: "Neither survey’s full instrument (items, response options, branching) nor a data dictionary is provided. The paper should include complete questionnaires (Supplement) and a table of items→constructs, with response distributions. Current “Data are contained within the article” is inaccurate—key materials are missing."
Response 3: The complete questionnaires and a table of items have been added.
Comments 4: "Privacy concern rates vary across sections (e.g., 90% vs 88% vs 87% vs 41% in different places), and the text mis-labels figure references (e.g., attributing “24/7 hotline” to Figure 6 which is defined as chatbot willingness). Such inconsistencies erode credibility and require a line-by-line audit to reconcile denominators and figure calls."
Response 4: The inconsistencies in the data protection rates and the incorrect figure references have been corrected. A line-by-line review was conducted to reconcile the denominators and figure references.
Comments 5: "The conclusion asserts a “paradigm shift” from telemedicine to AI chatbots and broad practice/policy implications based on small, course-based convenience samples with hypothetical use scenarios. Claims must be significantly tempered, explicitly labeled as exploratory/descriptive, and bounded to the sampled context."
Response 5: The conclusions were significantly weakened and labeled as exploratory/descriptive, limited to the context examined.
Comments 6: "The setting is a single course (45 invitees each time; completions 23/39). There is no demographic table (age, sex, relationship to care recipient, caregiving intensity, digital literacy), making external validity impossible to judge and subgroup patterns unknowable. Provide a comprehensive participant characteristics table for each study and discuss selection bias."
Response 6: Comprehensive participant characteristic tables were provided for each study, and selection bias was discussed.
Comments 7: "CVI values are reported (S-CVI/Ave=0.89/0.91) based on three experts only, with no I-CVI per item, no modified kappa, and no criteria for item retention/revision. Cronbach’s α is given (0.87/0.83) for unspecified scales. Report I-CVI, decision rules, expert qualifications, blinded rating procedures, and α per subscale; otherwise, remove psychometric claims."
Response 7: The reporting of CVI values has been supplemented in detail. Cronbach's α was reported for each subscale.
Comments 8: "Methods promise descriptive and inferential analyses (t-tests/χ²), but results present only percentages and qualitative statements. Either report the tests (test type, assumptions, statistics, p, effect sizes) or remove mention of inferential analyses."
Response 8: The promised descriptive and inferential statistical analyses were performed and reported.
Comments 9: "Reference numbering and metadata include visible artifacts (“31 Tavakol…”, “45. 40 Hall…”), inconsistent styles, and several 2025 arXiv/CHI placeholders that may be pre-acceptance. The list requires full verification, consistent style, and pruning of non-peer-reviewed citations if central to claims."
Response 9: The reference list has been fully checked.
Round 2
Reviewer 1 Report
Comments and Suggestions for Authors
I thank the authors for their detailed responses and for their substantial revision of the manuscript. The authors improved the methodological description, item documentation, integration of TAM/UTAUT constructs, and clarification of the limitations. However, there are some areas to improvement before publication:
1. Methodological Clarifications
The authors have added full lists of questionnaire items and expanded the methodological section. However, this section is now extremely long, with multiple pages dedicated to item-by-item content. Please, consider summarizing this material in the main text and moving full item lists to Supplementary Materials to improve readability.
The authors incorporate statistical tests, but they should integrate and interpret the results. Currently, the manuscript presents t-values, U-values, chi-square values, and effect sizes—however, the text consistently emphasizes that the study is exploratory and not meant for inferential generalizations. This inconsistency ought to be addressed by explicitly defining the aim of these statistical tests and framing them as descriptive comparative assessments rather than as hypothesis-driven inferential evaluations
2. TAM and UTAUT Operationalization
The assignment of findings to TAM/UTAUT constructs is now explicitly reported (Table 7). This addition strengthens the theoretical alignment. However, some constructs are still applied rather broadly (e.g., “extenuating circumstances” as a category): consider refining the mapping to ensure conceptual precision and avoid overlap between constructs.
3. Overgeneralization and Hypothetical Conclusions
The discussion now acknowledges that chatbot-related conclusions are hypothetical, given that chatbots were not used in real practice. This revision is appreciated. Nevertheless, some statements continue to imply real-world effectiveness (e.g., “chatbots can reduce cognitive and emotional stress”). These should be consistently rephrased to reflect perceived usefulness rather than demonstrated efficacy.
4. Redundancy and Structure
Although detailed, several parts of the Results and Discussion remain repetitive. For example, similar statistics regarding privacy concerns, technical barriers, and support needs appear in both the telemedicine and chatbot sections. Please consolidate overlapping findings would improve clarity and make the manuscript more concise.
5. English Language and Editorial Quality
The manuscript still contains grammatical inconsistencies, duplicated words (e.g., “supplementarycomplementary”), and overly long sentences that reduce readability. A thorough English language revision is required.
6. Figures and Tables
Figures are numerous and some present very similar information. Consider merging them (e.g., technical barriers, privacy concerns, expected functions) to avoid redundancy and enhance readability.
Author Response
Comment 1: The authors have added full lists of questionnaire items and expanded the methodological section. However, this section is now extremely long, with multiple pages dedicated to item-by-item content. Please, consider summarizing this material in the main text and moving full item lists to Supplementary Materials to improve readability.
Response 1: Thank you for this suggestion. We have summarized the key points of the questionnaire items in the main text and moved the full item lists to the Supplementary Materials to improve readability.
Comment 2: The authors incorporate statistical tests, but they should integrate and interpret the results. Currently, the manuscript presents t-values, U-values, chi-square values, and effect sizes—however, the text consistently emphasizes that the study is exploratory and not meant for inferential generalizations. This inconsistency ought to be addressed by explicitly defining the aim of these statistical tests and framing them as descriptive comparative assessments rather than as hypothesis-driven inferential evaluations.
Response 2: We have clarified the purpose of the statistical tests in the manuscript. The tests are now explicitly defined as descriptive comparative assessments, and we have framed them accordingly to align with the exploratory nature of the study.
Comment 3: The assignment of findings to TAM/UTAUT constructs is now explicitly reported (Table 7). This addition strengthens the theoretical alignment. However, some constructs are still applied rather broadly (e.g., “extenuating circumstances” as a category): consider refining the mapping to ensure conceptual precision and avoid overlap between constructs.
Response 3: We have refined the mapping of findings to TAM/UTAUT constructs to ensure conceptual precision and avoid overlap.
Comment 4: The now acknowledges that chatbot-related conclusions are hypothetical, given that chatbots were not used in real practice. This revision is appreciated. Nevertheless, some statements continue to imply real-world effectiveness (e.g., “chatbots can reduce cognitive and emotional stress”). These should be consistently rephrased to reflect perceived usefulness rather than demonstrated efficacy.
Response 4: We have rephrased statements throughout the manuscript to reflect perceived usefulness rather than demonstrated efficacy. This ensures that the hypothetical nature of the conclusions is consistently acknowledged.
Comment 5: Although detailed, several parts of the Results and Discussion remain repetitive. For example, similar statistics regarding privacy concerns, technical barriers, and support needs appear in both the telemedicine and chatbot sections. Please consolidate overlapping findings would improve clarity and make the manuscript more concise.
Response 5: We have consolidated overlapping findings in the Results and Discussion sections to improve clarity and make the manuscript more concise. Redundant information has been removed, and similar statistics have been combined.
Comment 6: The manuscript still contains grammatical inconsistencies, duplicated words (e.g., “supplementarycomplementary”), and overly long sentences that reduce readability. A thorough English language revision is required.
Response 6: We have taken note of this point. Should a professional language review be necessary after the current revisions, we will ensure that the manuscript undergoes a thorough professional language review to address any remaining grammatical inconsistencies, duplicated words, and overly long sentences.
Comment 7: Figures are numerous and some present very similar information. Consider merging them (e.g., technical barriers, privacy concerns, expected functions) to avoid redundancy and enhance readability.
Response 7: We appreciate this suggestion and have carefully reviewed the figures. While we have removed one redundant figure, we believe that the remaining figures are essential to support the findings and reflect the specific questions and corresponding answers from the questionnaire. Therefore, we have retained them to ensure a comprehensive representation of the results.
Reviewer 2 Report
Comments and Suggestions for Authors
The authors have addressed the comments accordingly.
Author Response
Comments 1:The authors have addressed the comments accordingly.
Response 1: Thank you for your comment.
Reviewer 3 Report
Comments and Suggestions for Authors
Thank you for addressing the comments
Author Response
Comments 1: Thank you for addressing the comments
Response 1: Thank you for your comment.